# HbA1c is inversely associated with thyroid cysts in a euthyroid population: A cross-sectional study

Yuji Shimizu[1,2]*, Shin-Ya Kawashiri[1], Yuko Noguchi[1], Yasuhiro Nagata[3], Takahiro Maeda[1,4], Naomi Hayashida[5]

1 Department of Community Medicine, Nagasaki University Graduate School of Biomedical Sciences, Nagasaki, Japan, 2 Department of Cardiovascular Disease Prevention, Osaka Center for Cancer and Cardiovascular Diseases Prevention, Osaka, Japan, 3 Center for Comprehensive Community Care Education, Nagasaki University Graduate School of Biomedical Sciences, Nagasaki, Japan, 4 Department of General Medicine, Nagasaki University Graduate School of Biomedical Sciences, Nagasaki, Japan, 5 Division of Promotion of Collaborative Research on Radiation and Environment Health Effects, Atomic Bomb Disease Institute, Nagasaki University, Nagasaki, Japan

* shimizu@osaka-ganjun.jp

**Data Availability Statement:** We cannot publicly provide individual data due to participant privacy, according to ethical guidelines in Japan. Additionally, the informed consent was obtained

## Abstract

Anti-thyroid peroxidase antibody (TPO-Ab) is revealed to be inversely associated with thyroid cysts among euthyroid population. TPO-Ab causes autoimmune thyroiditis by bolstering thyroid inflammation. Therefore, at least partly, absence of thyroid cysts could indicate latent thyroid damage. Since participants with subclinical hypothyroidism are reported to have higher HbA1c than normal healthy controls, HbA1c could be inversely associated with thyroid cysts through a mechanism reflecting latent thyroid damage. To investigate the association between HbA1c and thyroid cysts among a euthyroid population, a cross-sectional study was conducted on 1,724 Japanese individuals who were within the normal range of thyroid function [i.e., normal range of free triiodothyronine (T3) and free thyroxine (T4)] and aged 40–74 years. Among this study population, 564 were diagnosed with thyroid cysts. Independently of thyroid related hormones [thyroid stimulating hormone (TSH), free T3, and free T4] and known cardiovascular risk factors, HbA1c was found to be significantly inversely associated with the presence of thyroid cysts. This association remained significant even after this analysis was limited to participants within a normal range of TSH. The fully adjusted odds ratios (ORs) of thyroid cysts for 1 standard deviation (SD) increment of HbA1c were 0.84 (0.74, 0.95) for total participants and 0.80 (0.70, 0.92) for participants within a normal range of TSH. Among participants with normal thyroid function, HbA1c was inversely associated with the presence of thyroid cysts. The absence of thyroid cysts and higher levels of HbA1c could indicate the latent functional damage of the thyroid.

does not include a provision for publicity sharing data. Qualifying researchers may apply to access a minimal dataset by contacting Prof Naomi Hayashida, Principal Investigator, Division of Promotion of Collaborative Research on Radiation and Environment Health Effects, Atomic Bomb Disease Institute, Nagasaki University, Nagasaki, Japan at naomin@nagasaki-u.ac.jp. Or, please contact the office of data management at ritouken@vc.fctv-net.jp. Information for where data request is also available at https://www.genken. nagasaki-u.ac.jp/dscr/message/ and http://www. med.nagasaki-u.ac.jp/cm/.

**Funding:** This study was supported by Grants-in Aids for Scientific Research from Japan Society for the Promotion of Science (No. 17H03740 for TM, No. 17K09088 for NH, No. 18K06448 for YS). https://research-er.jp/categories/512 The funders had no role in study design, data collection and analysis, decision to publish, or preparation of the manuscript.

**Competing interests:** The authors have declared that no competing interests exist.

## Introduction

The fluid collected from thyroid cysts is reportedly rich in thyroglobulin [1], which contributes to the synthesis of thyroid hormones such as triiodothyronine (T3) and thyroxine (T4) [2].

We previously identified a significant positive association between thyroid cysts and isolated systolic hypertension among euthyroid participants [3]. Since hyperthyroidism is a major secondary cause of isolated systolic hypertension [4], euthyroid participants with thyroid cysts might have relatively higher thyroid hormone synthesizing activity than those without thyroid cysts.

In addition, our previous study reported an inverse association between anti-thyroid peroxidase antibody (TPO-Ab) and thyroid cysts among participants with normal thyroid hormone levels [5]. As TPO-Ab inhibits thyroid peroxidase, which plays an important role in synthesizing thyroid hormones [6], the absence of thyroid cysts in euthyroid individuals could indicate latent damage to the thyroid.

Another cross-sectional study reported that the levels of HbA1c in non-diabetic participants with subclinical hypothyroidism were significantly higher than those in normal healthy controls [7].

As the absence of thyroid cysts might indicate latent damage to the thyroid [5] and a lower productivity of thyroid hormone [8] and subclinical hypothyroidism might cause HbA1c elevation [7], HbA1c could be inversely associated with thyroid cysts among participants with normal thyroid function. However, no study has reported these associations.

To clarify these associations, we conducted a cross-sectional study of 1,724 Japanese individuals who participated in an annual health check-up in 2014 and who had normal range of thyroid function (i.e., normal range of free T3 and free T4) and aged 40–74 years.

## Materials and methods

### Study population

The methods that relate to the present risk surveys, including thyroid function, have also been described elsewhere [3,5,8].

Written consent forms were made available to ensure that participants understood the objective of the study, for which informed consent was obtained. This study was approved by the Ethics Committee of Nagasaki University Graduate School of Biomedical Sciences (project registration number 14051404).

The study population comprised 1,883 Japanese individuals aged 40–74 years from Saza town in the western part of Japan who underwent an annual medical check-up in 2014, as recommended by the Japanese government.

To preclude any influence of thyroid disease, participants with a history of thyroid disease (n = 60); without thyroid function data such as TSH, free T3, and free T4 (n = 17); and with abnormal ranges of free T3 (normal range: 2.1–4.1. pg/mL) and free T4 (normal range: 1.0–1.7. ng/dL) were excluded (n = 77).

Additionally, participants without data on body mass index (BMI) (n = 1), blood pressure (n = 1), or personal habits (smoking status and drinking status) (n = 3) were excluded.

The remaining participants, comprising 1,724 individuals with a mean age of 60.5 years [standard deviation (SD): 9.1; range: 40–74 years], were enrolled in the study.

### Data collection and laboratory measurements

A trained interviewer obtained information on clinical characteristics, such as history of thyroid disease, glucose-lowering medication use, and status of preferences (smoking and

drinking). Body weight and height were measured using an automatic body composition analyzer (BF-220; Tanita, Tokyo, Japan), and BMI ($kg/m^2$) was calculated. Systolic blood pressure (SBP) was recorded at rest.

Fasting blood samples were collected. TSH, free T3, and free T4 levels were measured by standard procedures at the LSI Medience Corporation (Tokyo, Japan). HbA1c, triglyceride (TG), and high-density lipoprotein cholesterol (HDLc) levels were also measured using standard procedures at SRL Inc. (Tokyo, Japan). The threshold for subclinical hypothyroidism is set at TSH > 4.01 μIU/mL.

The presence or absence of thyroid cysts was determined by experienced technicians using LOGIQ Book XP with a 10-MHz transducer (GE Healthcare, Milwaukee, WI, USA). A thyroid cyst was defined to have a maximum diameter of $\geq$2.0 mm without a solid component, as indicated in our previous study [3,5,8].

### Statistical analysis

Characteristics of the entire study population as well as cohorts divided by HbA1c levels (<5.6% for [low], 5.6–6.4% for [medium], and 6.5% $\leq$ for [high]) were expressed as means ± standard deviations (SDs), except for the categories of male sex, use of glucose-lowering medication, preference history (current smoker and daily drinker), and TSH. These categories were expressed as percentages. As TSH showed a skewed distribution, it was expressed as the median [first quartile, third quartile]. Significant differences in HbA1c levels were evaluated using analysis of variance.

Odds ratios (ORs) and 95% confidence intervals (CIs) were calculated using logistic regression models to assess association between HbA1c and subclinical hypothyroidism.

Simple correlation analysis was performed to evaluate the correlations between HbA1c and TSH levels and thyroid hormone (free T3 and free T4) levels.

Logistic regression models were also used to calculate ORs and 95% CIs to determine the association between HbA1c and thyroid cysts.

Three adjustment models were used. The first model was adjusted for sex and age (model 1). The second (model 2) further included the potential confounding factors that were directly associated with thyroid function, namely free T3 (pg/mL) and TSH (μIU/mL). The last model (model 3) was further adjusted for potential confounding factors that were indirectly associated with thyroid function, such as BMI ($kg/m^2$), smoking status (never, former, current), drinking status (non, often, daily), SBP (mmHg), TG level (mg/dL), and HDLc level (mg/dL). We also limited those analyses to participants with normal ranges of TSH and those not using glucose-lowering medication.

All statistical analyses were performed using the SAS system for Windows (version 9.4; SAS Inc., Cary, NC, USA). Results with p<0.05 were considered statistically significant.

## Results

In the study population, 564 (32.7%) were diagnosed to have thyroid cysts.

### Characteristics of the study population

Characteristics of the study population are shown in Table 1. Among the 1,724 patients enrolled, 37.0% were men. Free T3 and free T4 levels in the population were 3.17 ± 0.32 pg/mL and 1.25 ± 0.16 ng/dL, respectively. The median [the first quartile, the third quartile] levels of TSH were 1.58 [1.08, 2.29] μIU/mL.

**Table 1. Characteristics of the study population.**

| | |
|---|---|
| No of participants | 1,724 |
| Men, % | 37.0 |
| Age, year | 60.5 ± 9.1 |
| Free T3, pg/mL | 3.17 ± 0.32 |
| Free T4, ng/dL | 1.25 ± 0.16 |
| TSH, μIU/mL | 1.58 [1.08, 2.29][*1] |
| BMI, kg/m$^2$ | 22.8 ± 3.4 |
| SBP, mmHg | 125 ± 17 |
| Current smoker, % | 13.7 |
| Daily drinker, % | 40.3 |
| Glucose-lowering medication use, % | 6.1 |
| TG, mg/dL | 105 ± 74 |
| HDLc, mg/dL | 60 ± 15 |

T3; triiodothyronine, T4; thyroxine, TSH; thyroid stimulating hormone, BMI; body mass index, SBP; systolic blood pressure, TG; triglyceride, HDLc; high-density lipoprotein cholesterol. Values are means ± standard deviations.

[*1]: Values are median [the first quartile, the third quartile].

## Characteristics of the study population according to HbA1c levels

Table 2 shows characteristics of the study population based on HbA1c levels. Male gender, age, TSH, BMI, SBP, use of glucose-lowering medication, and TG were significantly and positively associated with HbA1c levels.

## Association between subclinical hypothyroidism and HbA1c

Table 3 shows a significant positive association between HbA1c and subclinical hypothyroidism that is independent of known cardiovascular risk factors.

**Table 2. Characteristics of study population according to HbA1c levels.**

| | HbA1c levels | | | |
|---|---|---|---|---|
| | < 5.6% [low] | 5.6–6.4% [medium] | 6.5%≤ [high] | P |
| No of participants | 985 | 616 | 123 | |
| Men, % | 36.9 | 34.4 | 51.2 | 0.002 |
| Age, year | 58.5 ± 9.5 | 63.0 ± 7.9 | 64.0 ± 7.2 | <0.001 |
| free T3, pg/mL | 3.17 ± 0.32 | 3.18 ± 0.32 | 3.13 ± 0.34 | 0.234 |
| free T4, ng/dL | 1.25 ± 0.16 | 1.24 ± 0.16 | 1.27 ± 0.17 | 0.400 |
| TSH, μIU/mL | 1.54 [1.05, 2.20][*1] | 1.60 [1.13, 2.39][*1] | 1.81 [1.20, 2.72][*1] | <0.001[*2] |
| BMI, kg/m$^2$ | 22.1 ± 3.1 | 23.4 ± 3.4 | 25.0 ± 3.9 | <0.001 |
| SBP, mmHg | 122 ± 16 | 127 ± 17 | 131 ± 18 | <0.001 |
| Current smoker, % | 14.9 | 10.6 | 19.5 | 0.069 |
| Daily drinker, % | 43.4 | 35.7 | 38.2 | 0.008 |
| Glucose-lowering medication use, % | 0.2 | 5.2 | 57.2 | <0.001 |
| TG, mg/dL | 97 ± 65 | 113 ± 78 | 138 ± 98 | <0.001 |
| HDLc, mg/dL | 62 ± 15 | 59 ± 14 | 54 ± 15 | <0.001 |

T3; triiodothyronine, T4; thyroxine, TSH; thyroid stimulating hormone, BMI; body mass index, SBP; systolic blood pressure, TG; triglycerides, HDLc; high-density lipoprotein cholesterol. Values are mean ±standard deviation.

[*1]: Values are median [the first quartile, the third quartile]. Regression model for mean values was used for determining p values.

[*2]: Logarithmic transformation was used for evaluating p.

**Table 3. Odds ratios (ORs) and 95% confidence intervals (CIs) of subclinical hypothyroidism in relation to HbA1c levels.**

| | HbA1c levels | | | p | 1 SD increment of HbA1c |
|---|---|---|---|---|---|
| | < 5.6% [low] | 5.6–6.4% [medium] | 6.5%≤ [high] | | |
| No of participants | 985 | 616 | 123 | | |
| No. of case (%) | 36 (3.7) | 43 (7.0) | 19 (15.4) | | |
| Model 1 | 1 | 1.98 (1.24, 3.16) | 2.39 (1.34, 4.29) | <0.001 | 1.44 (1.35, 1.66) |
| Model 2 | 1 | 2.01 (1.26, 3.21) | 2.27 (1.26, 4.09) | <0.001 | 1.43 (1.19, 1.71) |
| Model 3 | 1 | 1.74 (1.08, 2.82) | 2.03 (1.12, 3.69) | 0.003 | 1.35 (1.16, 1.58) |

Case: Participants with subclinical hypothyroidism. Model 1: Adjusted for sex and age. Model 2: + free T3 and TSH. Model 3: + BMI, smoking status (never, former, current), drinking status (non, often, daily), SBP, TG and HDLc. 1 standard deviation (SD) increment of HbA1c was 0.7% for men and 0.6% for women.

## Correlation among HbA1c, TSH, and thyroid hormone

Table 4 shows the correlation among HbA1c, TSH, and thyroid hormone levels. HbA1c was significantly and positively correlated with TSH level in all participants but not in those within normal TSH range. For both groups, thyroid hormones (free T3 and free T4) showed no significant association with HbA1c level but were significantly inversely associated with TSH level.

## Association between thyroid cysts and HbA1c levels in all participants and in those within normal TSH range

Table 5 shows the association between thyroid cysts and HbA1c levels for all participants and those within the normal range of TSH. HbA1c level was significantly inversely associated with thyroid cysts for both total participants and those within the normal range of TSH. These associations remained significant even after further adjustment for thyroid function and known cardiovascular risk factors.

## Association between thyroid cysts and HbA1c levels among participants not receiving glucose-lowering medication

As glucose-lowering medications might have influenced our study results, we also conducted an analysis limited to participants not receiving such medications, as shown in Table 6.

**Table 4. Correlation among HbA1c, thyroid stimulating hormone (TSH) and thyroid hormones (free T3 and free T4) levels.**

| | TSH | | Triiodothyronine (free T3) | | Thyroxine (free T4) | |
|---|---|---|---|---|---|---|
| | r | (p) | R | (p) | r | (p) |
| **Total** | | | | | | |
| No of participants | 1724 | | | | | |
| HbA1c | 0.11 | <0.001 | -0.04 | 0.074 | 0.01 | 0.605 |
| TSH | - | - | -0.12 | <0.001 | -0.15 | <0.001 |
| **Normal TSH** | | | | | | |
| No of participants | 1598 | | | | | |
| HbA1c | 0.05 | 0.069 | -0.03 | 0.217 | 0.04 | 0.143 |
| TSH | - | - | -0.10 | <0.001 | -0.08 | <0.001 |

r: Simple correlation coefficient.

**Table 5. Odds ratios (ORs) and 95% confidence intervals (CIs) of thyroid cyst in relation to HbA1c levels.**

| | HbA1c levels | | | p | 1 SD increment of HbA1c |
|---|---|---|---|---|---|
| | < 5.6% [low] | 5.6–6.4% [medium] | 6.5%≤ [high] | | |
| **Total** | | | | | |
| No of participants | 985 | 616 | 123 | | |
| No. of case (%) | 320 (32.5) | 216 (35.1) | 28 (22.8) | | |
| Model 1 | 1 | 0.95 (0.76, 1.19) | 0.57 (0.36, 0.91) | 0.041 | 0.83 (0.74, 0.94) |
| Model 2 | 1 | 0.96 (0.77, 1.20) | 0.57 (0.36, 0.91) | 0.046 | 0.83 (0.74, 0.94) |
| Model 3 | 1 | 0.97 (0.77, 1.22) | 0.58 (0.36, 0.92) | 0.075 | 0.84 (0.74, 0.95) |
| **Normal range of TSH** | | | | | |
| No of participants | 930 | 565 | 103 | | |
| No. of case (%) | 303 (32.6) | 202 (35.8) | 21 (20.4) | | |
| Model 1 | 1 | 0.97 (0.77, 1.22) | 0.48 (0.29, 0.80) | 0.035 | 0.80 (0.71, 0.92) |
| Model 2 | 1 | 0.97 (0.77, 1.23) | 0.48 (0.28, 0.80) | 0.035 | 0.80 (0.71, 0.92) |
| Model 3 | 1 | 0.98 (0.77, 1.23) | 0.48 (0.28, 0.81) | 0.045 | 0.80 (0.70, 0.92) |

Case: Participants with thyroid cysts. Model 1: Adjusted for sex and age. Model 2: + free T3 and TSH. Model 3: + BMI, smoking status (never, former, current), drinking status (non, often, daily), SBP, TG and HDLc. 1 standard deviation (SD) increment of HbA1c were 0.7% for men and 0.6% for women.

Essentially, the same associations were noted for both total participants and participants with normal TSH range.

## Sex-specific analysis among participants with in normal range of TSH and not receiving glucose-lowering medication

For sensitivity analysis, we also performed sex-specific analysis among participants not receiving glucose-lowering medication and found essentially the same associations. Fully adjusted ORs and 95%CIs of thyroid cysts for 1 SD increment of HbA1c were 0.87 (0.66, 1.16) for men (n [euthyroid individuals] = 536, cases [thyroid cysts] = 139) and 0.74 (0.59, 0.99) for women (n = 971, cases = 361), respectively.

**Table 6. Odds ratios (ORs) and 95% confidence intervals (CIs) of thyroid cyst in relation to HbA1c levels limited to participants not receiving glucose-lowering medication.**

| | HbA1c levels | | | p | 1 SD increment of HbA1c |
|---|---|---|---|---|---|
| | < 5.6% [low] | 5.6–6.4% [medium] | 6.5%≤ [high] | | |
| **Total** | | | | | |
| No of participants | 983 | 584 | 52 | | |
| No. of case (%) | 319 (32.5) | 205 (35.1) | 10 (19.2) | | |
| Model 1 | 1 | 0.96 (0.72, 1.20) | 0.46 (0.23, 0.95) | 0.146 | 0.83 (0.71, 0.97) |
| Model 2 | 1 | 0.97 (0.77, 1.21) | 0.46 (0.23, 0.95) | 0.161 | 0.84 (0.71, 0.98) |
| Model 3 | 1 | 0.98 (0.78, 1.23) | 0.48 (0.23, 1.00) | 0.249 | 0.85 (0.71, 0.997) |
| **Normal range of TSH** | | | | | |
| No of participants | 928 | 535 | 44 | | |
| No. of case (%) | 302 (32.5) | 191 (35.7) | 7 (15.9) | | |
| Model 1 | 1 | 0.98 (0.77, 1.23) | 0.36 (0.16, 0.84) | 0.148 | 0.79 (0.67, 0.94) |
| Model 2 | 1 | 0.98 (0.72, 1.23) | 0.36 (0.16, 0.84) | 0.151 | 0.79 (0.67, 0.94) |
| Model 3 | 1 | 0.99 (0.78, 1.25) | 0.38 (0.16, 0.88) | 0.222 | 0.80 (0.67, 0.95) |

Case: Participants with thyroid cysts. Model 1: Adjusted for sex and age. Model 2: + free T3 and TSH. Model 3: + BMI, smoking status (never, former, current), drinking status (non, often, daily), SBP, TG and HDLc. 1 standard deviation (SD) increment of HbA1c were 0.7% for men and 0.6% for women.

## Discussion

The major findings of the present study are that, independent of thyroid function, HbA1c is significantly inversely associated with thyroid cysts in the euthyroid population. This significant association remained even after the analyses were limited to participants with a normal range of TSH.

The absence of thyroid cysts might indicate latent damage to the thyroid, as a previous study on a euthyroid population revealed an independent inverse association between thyroid cysts and TPO-Ab, which is a known cause of autoimmune thyroiditis [5].

A high prevalence of TPO-Ab (+) has been reported in subclinical hypothyroidism [9,10], and hypothyroidism is known to be associated with hypertension [11]. Our previous study also revealed a positive association between subclinical hypothyroidism and hypertension among participants without thyroid cysts but not among participants with thyroid cysts [3]. Thus, the absence of thyroid cysts might have a disadvantage in the production of thyroid hormones, possibly by indicating latent damage of the thyroid.

In the present study, we found further evidence that HbA1c was significantly inversely associated with thyroid cysts in the euthyroid population (Tables 5 and 6).

Both hyperthyroidism and hypothyroidism are reported to be associated with the development of insulin resistance [12]. However, a positive linear association between the normal range of TSH level and insulin resistance among non-diabetes participants and type 2 diabetes patients has been reported [13]. Furthermore, a positive correlation between HbA1c and TSH levels was also observed in patients with subclinical hypothyroidism [14].

Hypothyroidism can break the equilibrium between thyroid hormone and glucose homeostasis and alter glucose metabolism, which can lead to insulin resistance [15]. Insulin resistance elevates HbA1c levels, and higher HbA1c levels could be associated with lower thyroid hormone activity, partly by indicating latent damage to the thyroid. Generally, levels of TSH that are high but within the normal range of thyroid hormones (free T3 and free T4) are considered to indicate subclinical hypothyroidism.

In the present study, we found a significant positive association between subclinical hypothyroidism and HbA1c (Table 3). This is consistent with the findings of a previous study that reported the levels of HbA1c in non-diabetic participants with subclinical hypothyroidism to be higher than those in normal healthy controls [7].

As the absence of thyroid cysts might indicate latent damage to the thyroid [3,5,8], insulin resistance related to TSH levels might underlie the association between thyroid cysts and HbA1c. In fact, in our additional analysis, a slight but significant positive association between TSH and HbA1c levels was observed, while significant inverse associations between TSH and thyroid hormone (free T3 and free T4) levels were observed among all euthyroid participants (Table 4).

However, the magnitude of the observed associations (between TSH and HbA1c, and between TSH and thyroid hormones) were too small to explain the present results showing a significant inverse association between HbA1c and thyroid cysts. The magnitude of physiological demand for thyroid hormone activity might explain this discrepancy. Because decreased thyroid function may extend longevity, the demand for thyroid hormone activity may decrease with age [16]. Therefore, not only the serum thyroid hormone level, but also the demand for thyroid activity might determine the level of TSH secretion. HbA1c levels may indicate the magnitude of thyroid hormone deficiency in the same way that the equilibrium between thyroid hormone and glucose might determine insulin resistance [15]. In the present study, HbA1c was significantly positively correlated with subclinical hypothyroidism (Table 3) but not with thyroid hormone (Table 4). Because the absence of thyroid cysts indicates latent damage to the thyroid [3,5,8], HbA1c levels could inversely associate with thyroid cysts (Table 5).

The same associations between HbA1c levels and thyroid cysts were observed when analyses were limited to the normal TSH range (Table 5).

Therefore, the absence of thyroid cysts and higher levels of HbA1c could indicate the latent functional damage of the thyroid.

Because the same glucose-lowering medication could influence thyroid function [17], which could influence the main results of our study, we conducted further analyses limited to participants who were not receiving glucose-lowering medications. A significant inverse association between HbA1c levels and thyroid cysts was observed even in the analyses that were limited to participants not receiving glucose-lowering medications (Table 6). Therefore, glucose-lowering medications that influence HbA1c levels could not explain the present main findings. Further investigation that directly indicates insulin resistance, such as HOMA-IR, is necessary.

Insulin resistance is known to be associated with thyroid function [12,13]. Thus, the present finding showing an inverse association between HbA1c level and thyroid cysts among participants with completely normal thyroid function (normal range of TSH and thyroid hormones such as free T3 and free T4) could indicate that thyroid cysts act as an indicator of thyroid activity, which could not be evaluated based on TSH, free T3, and free T4 levels. Therefore, the present study provides new insights for evaluating thyroid function in general clinical practice, although thyroid cysts are considered to lack significance in general. Furthermore, the importance of HbA1c level control in patients with subclinical hypothyroidism has been emphasized in a previous study [14]. Present findings could provide efficient cues to control HbA1c associated with the latent functional damage of the thyroid.

The potential limitations of this study warrant consideration. First, we evaluated the existence of thyroid cysts solely by their presence or absence. However, the volumes and number of cysts might also influence the present results, and further investigations using these data are necessary. Second, this was a cross-sectional study, and causal relationships could not be established. Third, the turnover of thyroid hormone may be influenced by status of thyroid cysts, but we could not evaluate the half-life of thyroid hormone.

## Conclusion

In conclusion, among participants with normal thyroid function, HbA1c level was found to be inversely associated with the presence of thyroid cysts. Although further investigations on the present topic are necessary, the absence of thyroid cysts and higher levels of HbA1c could indicate the latent functional damage of the thyroid.

## Acknowledgments

We are grateful to Ms. Keiko Yamaoka, Ms. Kaori Yamamura, and staff from Saza town office for their outstanding support.

## Author Contributions

**Conceptualization:** Yuji Shimizu, Takahiro Maeda, Naomi Hayashida.

**Data curation:** Yuji Shimizu, Shin-Ya Kawashiri, Yuko Noguchi, Yasuhiro Nagata, Takahiro Maeda, Naomi Hayashida.

**Formal analysis:** Yuji Shimizu.

**Funding acquisition:** Yuji Shimizu, Takahiro Maeda, Naomi Hayashida.

**Investigation:** Yuji Shimizu, Yasuhiro Nagata, Takahiro Maeda.

**Methodology:** Yuji Shimizu, Takahiro Maeda, Naomi Hayashida.

**Project administration:** Yuji Shimizu, Takahiro Maeda, Naomi Hayashida.

**Resources:** Yuji Shimizu, Yasuhiro Nagata, Takahiro Maeda, Naomi Hayashida.

**Software:** Yuji Shimizu.

**Supervision:** Naomi Hayashida.

**Validation:** Yuji Shimizu, Yuko Noguchi, Takahiro Maeda, Naomi Hayashida.

**Visualization:** Yuji Shimizu, Naomi Hayashida.

**Writing – original draft:** Yuji Shimizu.

**Writing – review & editing:** Yuji Shimizu.

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
