## [Decision Letter · Decision Letter 0]

10 Dec 2020

PONE-D-20-13979

HbA1c is inversely associated with thyroid cyst among eu-thyroid population: a cross-sectional study

PLOS ONE

Dear Dr. Shimizu,

Thank you for submitting your manuscript to PLOS ONE. After careful consideration, we feel that it has merit but does not fully meet PLOS ONE’s publication criteria as it currently stands. Therefore, we invite you to submit a revised version of the manuscript that addresses the points raised during the review process.

We look forward to receiving your revised manuscript.

Kind regards,

Yoshihiro Kokubo, PhD, MD, FACC, FAHA, FESC, FESO

Academic Editor

PLOS ONE

Journal Requirements:

Reviewers' comments:

Reviewer's Responses to Questions

**Comments to the Author**

1. Is the manuscript technically sound, and do the data support the conclusions?

Reviewer #1: Partly

Reviewer #2: No

2. Has the statistical analysis been performed appropriately and rigorously? 

Reviewer #1: Yes

Reviewer #2: No

3. Have the authors made all data underlying the findings in their manuscript fully available?

Reviewer #1: No

Reviewer #2: No

4. Is the manuscript presented in an intelligible fashion and written in standard English?

Reviewer #1: No

Reviewer #2: No

5. Review Comments to the Author

Reviewer #1: The manuscript entitled “HbA1c is inversely associated with thyroid cyst among euthyroid population: a cross sectional study” describes an inverse association between HbA1c levels and thyroid cysts in euthyroid patients. The Authors interpret this finding as a potential tool to explore a reduction in thyroid function.

The idea is original but the manuscript leaves too many edges without a clear explanation.

Major comments

1. Introduction: page 6 line 103: without preference data. Do you mean personal habits?

2. The introduction could benefit of further description on how 2 apparently disconnected parameters such as HbA1c and TPOab are linked. Is the carbohydrate alteration related to thyroid autoimmunity ?

3. Considering that thyroid cysts are common and 2 mm is a small size it would be better to clarify what the Authors refer to with “status for cysts” is it “yes or no” or “number of cysts”?

4. In statistical methods it would be useful to describe the tests in relation to what is shown later in the results section. In table 1 patients are distributed in 3 groups according to HbA1c levels but this is not introduced in methods. For the logistic regression where does it say that HbA1c and cysts are going to be analysed as it states in table 3?

5. A description of the biochemical and anthropometric parameters of the whole population should be included.

6. Which is the proportion of cysts according to HbA1c tertiles?

7. Table 1 please define levels of low, median, high level of HbA1c

8. References are quite scarce considering the abundance on the topic of thyroid and carbohydrate metabolism.

9. The manuscript should be revised for the English language.

Minor comments

1. Several grammar mistakes have to be corrected as in lines 29-31-39-68-71-72-118-132-143-164-165-172-173-194-196-231-254

-

Reviewer #2: This manuscript suffers from many flaws, making it very difficult to read and evaluate. For example, there are several points inconsistent between the text and the tables, the quality of English is very poor, and the meaning of the conclusion is also unclear.

6. PLOS authors have the option to publish the peer review history of their article (what does this mean?). If published, this will include your full peer review and any attached files.

Reviewer #1: **Yes: **Gabriela Brenta

Reviewer #2: No

---

## [Author Response · Author response to Decision Letter 0]

20 Dec 2020

Reviewer #1: The manuscript entitled “HbA1c is inversely associated with thyroid cyst among euthyroid population: a cross sectional study” describes an inverse association between HbA1c levels and thyroid cysts in euthyroid patients. The Authors interpret this finding as a potential tool to explore a reduction in thyroid function.

The idea is original but the manuscript leaves too many edges without a clear explanation.

(Answer.1)

→Thank you for valuable comment. I thought the point that reader feel lack of clear explanation is in our basal hypothesis; absence of thyroid cysts might indicate latent damage of thyroid.

In addition to the fact that the fluid collected from thyroid cysts is reported to be rich in thyroglobulin [Ref1], at that time when I submit present manuscript, only one article which supports above mentioned hypothesis [Ref2]; anti-thyroid peroxidase antibody (TPO-Ab) shows inversely associated with thyroid cysts. Since TPO-Ab is well known cause of autoimmune thyroiditis, participants with higher titer of TPO-Ab might have latent damage in thyroid. And TPO-Ab is significantly inversely associated with thyroid cysts. Then we thought, absence of thyroid cyst might indicate latent damage of thyroid.

However, we published another article during submitting present manuscript that explains thyroid cyst could influence on thyroid function [Ref2]. And also we could publish the article that explain the potential mechanism that underlying the TPO-Ab and subclinical hypothyroidism [Ref3]. In this article, we reported that presence of thyroid cysts has influenced on the association between TPO-Ab and subclinical hypothyroidism. Therefore, we could better explanation now. Then we revised by using those our previous studies as references.

[Ref1]

Salabé GB, et al. Identification of serum proteins, thyroglobulin and antithyroid antibodies in the fluid of thyroid cysts. Thyroidology. 1990;2(1):17-23.

[Ref2] 

Shimizu Y, et al. Anti-thyroid peroxidase antibody and thyroid cysts among the general Japanese population: a cross-sectional study. Environ Health Prev Med. 2020;25(1):7.

[Ref3]

Shimizu Y, et al. Association between thyroid-stimulating hormone and hypertension according to thyroid cysts status in the general population: a cross-sectional study. Environ Health Prev Med. 2020;25(1):69. 

[Ref4]

Shimizu Y, et al. Anti-thyroid peroxidase antibody and subclinical hypothyroidism in relation to hypertension and thyroid cysts. PLoS One. 2020;15(10):e0240198.

Because of those reasons, I revised abstract as following. Those sentences explain functional role that might thyroid cyst possess.

 A stronger correlation between thyroid-stimulating hormone (TSH) and free triiodothyronine (T3) was reported among euthyroid participants without thyroid cysts than among those with thyroid cysts. This correlation could be enhanced by low productivity of thyroid hormones. Then, absence of thyroid cysts could indicate low activity of thyroid production.

And I also added the following sentences in introduction section that explain the functional role that might thyroid cyst could have.

 The correlation between thyroid-stimulating hormone (TSH) and free T3 might be enhanced among participants with low thyroid hormone productivity, which is related to latent damage to the thyroid. Previously, we reported that euthyroid individuals without thyroid cysts had a stronger correlation between TSH and free T3 than those with thyroid cysts [3]. Therefore, the absence of thyroid cysts might negatively affect thyroid hormone synthesis.

After describing the potential functional role that thyroid cysts might have, we describe the association between TPO-Ab and thyroid cysts because this association indicates that absence of thyroid cyst could indicate latent damage of thyroid.

And in discussion section, we first described about the latent damage of thyroid that relates to subclinical hypothyroidism. And clarified the association between TPO-Ab and hypothyroidism in relation to thyroid cysts as following.

A high prevalence of TPO-Ab (+) has been reported in subclinical hypothyroidism [8,9], and hypothyroidism is known to be associated with hypertension [10]. Our previous study also revealed a positive association between subclinical hypothyroidism and hypertension among participants without thyroid cysts but not among participants with thyroid cysts [7]. Thus, the absence of thyroid cysts might have a disadvantage in the production of thyroid hormones, possibly by indicating latent damage of the thyroid.

In the present study, we found further evidence that HbA1c was significantly inversely associated with thyroid cysts in the euthyroid population (Table 4).

Major comments

1. Introduction: page 6 line 103: without preference data. Do you mean personal habits?

(Answer.2)

→Thank you for valuable comment. According to this reviewer’s valuable comment, I revised as following.

 personal habits (smoking status and drinking status)

2. The introduction could benefit of further description on how 2 apparently disconnected parameters such as HbA1c and TPOab are linked. Is the carbohydrate alteration related to thyroid autoimmunity ?

(Answer.3)

→　

Thank you for valuable comment. We reconsider the meaning of anti-thyroid peroxidase antibody (TPO-Ab) in present study. Previously, we reported inverse association between TPO-Ab and thyroid cysts among eu-thyroid population [Ref2]. Since TPO-Ab is known cause of autoimmune thyroiditis that induce thyroid damage, we thought absence of TPO-Ab among eu-thyroid population could indicate latent damage of thyroid. Therefore, the fact that absence of thyroid cyst indicates latent damage of thyroid is important but not TPO-Ab. However, thanks to this reviewer’s valuable comment, I recognized that present description could mislead. Then we reconsider the way to describe the fact that absence of thyroid cyst might indicate latent thyroid damage that relates to lower productivity of thyroid hormone. 

Previously, we also reported that eu-thyroid participants without thyroid cysts shows stronger correlation between thyroid-stimulating hormone (TSH) and free triiodothyronine (T3) than that of with thyroid cysts [Ref02]. Since this correlation could be enhanced by low productivity of thyroid hormone, at least partly, absence of thyroid cysts could indicate low activity of thyroid production.

Then, to avoid the misleading, first, we focus on the thyroid cysts status specific association between TSH and free T3 which indicates that absence of thyroid cysts might have disadvantage in producing thyroid hormone. And after that, we also introduce our previous study that shows significant inverse association between thyroid cysts and TPO-Ab among eu-thyroid population because this inverse association also indicates that absence of thyroid cyst among eu-thyroid population might indicates latent damage of thyroid.

In addition to that, even absence of thyroid cysts might indicate latent damage of thyroid [Ref2], the potential influence of thyroid cysts on thyroid function is not yet discussed in introduction section. We also reported in previous study that indicates absence of thyroid cysts could be associated with lower function of thyroid [Ref4]. 

The specific change in present manuscripts were described for former answer (Answer.1).

3. Considering that thyroid cysts are common and 2 mm is a small size it would be better to clarify what the Authors refer to with “status for cysts” is it “yes or no” or “number of cysts”?

(Answer.4)

→Thank you for valuable comment. According to this reviewer’s valuable comment, after edited by native English check we revised as following.

The presence or absence of thyroid cysts was determined by experienced technicians using LOGIQ Book XP with a 10-MHz transducer (GE Healthcare, Milwaukee, WI, USA). A thyroid cyst was defined to have a maximum diameter of ≥2.0 mm without a solid component, as indicated in our previous study [3,4,7].

4. In statistical methods it would be useful to describe the tests in relation to what is shown later in the results section. In table 1 patients are distributed in 3 groups according to HbA1c levels but this is not introduced in methods. For the logistic regression where does it say that HbA1c and cysts are going to be analysed as it states in table 3?

(Answer.5)

→Thank you for valuable comment. According to this reviewer’s valuable comment, we described as following.

Characteristics of the entire study population as well as the entire population based on HbA1c levels (<5.4% for T1 [low], 5.4-5.6 % for T2 [medium], and 5.7% ≤ for T3 [high]) were expressed as means ± SDs, except for the categories of male sex, use of glucose-lowering medication, preference history (current smoker and daily drinker), and TSH. These categories were expressed as percentages. As TSH showed a skewed distribution, it was expressed as the median (first quartile, third quartile). Significant differences in HbA1c levels were evaluated using analysis of variance.

5. A description of the biochemical and anthropometric parameters of the whole population should be included.

(Answer.6)

→Thank you for valuable comment. According to this reviewer’s valuable comment, we added table 1 that shows biochemical and anthropologic parameter of the whole population. And we also added the following sentences in result section.

In the study population, 564 (32.7%) were diagnosed to have thyroid cysts.

Characteristics of the study population

 Characteristics of the study population are shown in Table 1. Among the 1,724 patients enrolled, 37.0 % were men. FT3 and FT4 levels in the population were 3.17 ± 0.32 pg/mL and 1.25 ± 0.16 ng/dL, respectively. The median [the first quartile, the third quartile] levels of TSH were 1.58 [1.08, 2.29] μIU/mL.

6. Which is the proportion of cysts according to HbA1c tertiles?

(Answer.7)

→Thank you for valuable comment. According to this reviewer’s valuable comment, we rechecked the present analyzes. Since our present study evaluates the association between HbA1c levels and thyroid cysts, the proportion of cysts according to HbA1c tertiles were shown in tables (Table 4 and Table 5) with description of “No. of cases”. To better 

To better understand, we added following sentence in foot note of table 4 and table 5.

Cases: participants with thyroid cysts.

7. Table 1 please define levels of low, median, high level of HbA1c

(Answer.8)

→Thank you for valuable comment. According to this reviewer’s valuable comment, we rechecked the table. Since we already described the exact number of HbA1c tertile levels in footnote of the tables. We added following sentence in first paragraph of the statistical analyzes section.

Characteristics of the entire study population as well as the entire population based on HbA1c levels (<5.4% for T1 [low], 5.4-5.6 % for T2 [medium], and 5.7% ≤ for T3 [high])

8. References are quite scarce considering the abundance on the topic of thyroid and carbohydrate metabolism.

(Answer.9)

→Thank you for valuable comment. As above mentioned reason, we could include recent our reports and strength the background of our hypothesis. In addition to that, thanks to this reviewer’s valuable advice we added following sentence.

Hypothyroidism can break the equilibrium between thyroid hormone and glucose homeostasis and alter glucose metabolism, which can lead to insulin resistance [14].

Then the number of references became almost double. 

9. The manuscript should be revised for the English language.

Minor comments

1. Several grammar mistakes have to be corrected as in lines 29-31-39-68-71-72-118-132-143-164-165-172-173-194-196-231-254

(Answer.10)

→Thank you for valuable comment. Even before submit this manuscript, our manuscript was edited by professional English editing service, I realized there are many grammatical mistake. I informed this issue to staff of professional English editing service and made them check whole sentences again.

Reviewer #2: This manuscript suffers from many flaws, making it very difficult to read and evaluate. For example, there are several points inconsistent between the text and the tables, the quality of English is very poor, and the meaning of the conclusion is also unclear.

 (Answer)

→　

Thank you for valuable comment. According to this reviewer’s valuable comment, I reconsider the content of present manuscript.

*To clarify the association between HbA1c and thyroid cyst among participants with normal thyroid function is the main purpose of present study because those findings indicates that even participants with latent thyroid damage showed normal range of thyroid hormone, those participants might have the same disadvantage in thyroid hormone activity.

However, before discuss about the above mentioned hypothesis, we have to clarify the fact that a) absence of thyroid cyst could indicate the latent damage of thyroid and b) elevated HbA1c levels could be observed among latent damage of thyroid.

a) About the absence of thyroid cyst as a latent damage of thyroid 

Previously, we reported that absence of thyroid cysts could act as an indicator of latent thyroid damage by following three reasons (1~3).

1) Thyroid peroxidase antibody (TPO-Ab) which is well known cause of auto immune thyroiditis is reveled to be significantly inversely associated with thyroid cyst among participants with normal thyroid function [Ref1]. Since TPO-Ab might induce thyroid damage even among participants with normal thyroid function, absence of thyroid cyst could indicate latent damage of thyroid.

[Ref1]

Shimizu Y, et al. Anti-thyroid peroxidase antibody and thyroid cysts among the general Japanese population: a cross-sectional study. Environ Helath Prev Med. 2020;25(1):7.

2) In addition to above mentioned reasons, we also found significant positive association between subclinical hypothyroidism and hypertension among subjects without thyroid cysts but not among subjects with thyroid cysts [Ref2]. Since high prevalence of TPO-Ab is reported in subclinical hypothyroidism [Ref3][Ref4] and hypothyroidism is known to be associated with hypertension [Ref5], absence of thyroid cyst also might have disadvantage in production of thyroid hormone.

[Ref2]

Shimizu T, et al. Anti-thyroid peroxidase antibody and subclinical hypothyroidism in relation to hypertension and thyroid cysts. PLoS One. 2020;15(10):e0240198.

[Ref3]

Darya S, et al. Assessment of subclinical hypothyroidism for a clinical score and thyroid peroxidase antibody: a comparison with euthyroidism grouped by different thyroid-stimulating hormone levels. Asian Biomedicine. 2019;13:85-93.

[Ref4]

Mohanty S, et al. Diagnostic strategies for subclinical hypothyroidism. Indian J Clin Biochem. 2008;23(3):279-282.

[Ref5]

 Stabouli S, et al. Hypothyroidism and hypertension. Expert Rev Cardiovasc Ther. 2010;8(11):1559-1565.

3) Furthermore, previously, we found that among eu-thyroid population, subjects without thyroid cysts shows stronger correlation between thyroid stimulating hormone (TSH) and thyroid hormone than that of with thyroid cysts [Ref6]. Since the correlation between thyroid-stimulating hormone (TSH) and free T3 might be enhanced among participants with low thyroid hormone productivity that relates to latent damage of thyroid, absence of thyroid cysts could indicate latent damage of thyroid.

[Ref6]

Shimizu Y, et al. Association between thyroid-stimulating hormone and hypertension according to thyroid cysts status in the general population: a cross-sectional study. Environ Helath Prev Med. 2020;25(1):69.

b) About elevated HbA1c levels could be observed among latent damage of thyroid.

Hypothyroidism can break the equilibrium between thyroid hormone and glucose homeostasis and alter glucose metabolism which can lead to insulin resistance [Ref7] and insulin resistance elevates HbA1c level. Then higher HbA1c could be associated with lower activity of thyroid hormone partly by indicating latent damage of thyroid.

[Ref7]

Brenta G. Why can insulin resistance be a natural consequence of thyroid dysfunction? J Thyroid Res. 2011;2011:152850.

Since those manuscripts [Ref2] and [Ref6] were not published when I submit present manuscript, we could not include in text. However, now we can use those references then we revised.

In abstract, we as following that explaining the potential function that thyroid cyst might possess.

A stronger correlation between thyroid-stimulating hormone (TSH) and free triiodothyronine (T3) was reported among euthyroid participants without thyroid cysts than among those with thyroid cysts. This correlation could be enhanced by low productivity of thyroid hormones. Then, absence of thyroid cysts could indicate low activity of thyroid production.

And we added the following sentences in introduction section.

 The correlation between thyroid-stimulating hormone (TSH) and free T3 might be enhanced among participants with low thyroid hormone productivity, which is related to latent damage to the thyroid. Previously, we reported that euthyroid individuals without thyroid cysts had a stronger correlation between TSH and free T3 than those with thyroid cysts [3]. Therefore, the absence of thyroid cysts might negatively affect thyroid hormone synthesis. 

After describing above mentioned sentences, we described about the association between TPO-Ab and thyroid cysts because presence of TPO-Ab among participants with normal thyroid function could indicate presence of latent thyroid damage. Since subclinical hypothyroidism is one of the condition that is associated with thyroid damage, we described following sentences in discussion section. 

The absence of thyroid cysts might indicate latent damage to the thyroid, as a previous study on a euthyroid population revealed an independent inverse association between thyroid cysts and TPO-Ab, which is a known cause of autoimmune thyroiditis [4].

A high prevalence of TPO-Ab (+) has been reported in subclinical hypothyroidism [8,9], and hypothyroidism is known to be associated with hypertension [10]. Our previous study also revealed a positive association between subclinical hypothyroidism and hypertension among participants without thyroid cysts but not among participants with thyroid cysts [7]. Thus, the absence of thyroid cysts might have a disadvantage in the production of thyroid hormones, possibly by indicating latent damage of the thyroid.

In addition, we described as following sentences that indicates lower activity of thyroid could be associated with insulin resistance.

Hypothyroidism can break the equilibrium between thyroid hormone and glucose homeostasis and alter glucose metabolism, which can lead to insulin resistance [14].

*To clarify the association between description is text and tables, we added table number in discussion section where sentences is described present results. 

* We also added the following sentences that explain why we made further analyzes limited to participants without taking glucose lowering medication. And describe what can be indicating from these additional analyzes.

 Because the same glucose-lowering medication could influence thyroid function [15], which could influence the main results of our study, we conducted further analyses limited to participants who were not receiving glucose-lowering medications. A significant inverse association between HbA1c levels and thyroid cysts was observed even in the analyses that were limited to participants not receiving glucose-lowering medications (Table 5). Therefore, glucose-lowering medications that influence HbA1c levels could not explain the present main findings. 

*Our present manuscript has already edited by professional English editing service. Even though, same grammatical error was pointed. Then we informed this valuable comment to the professional English service and made them editing for all manuscript again.

---

## [Decision Letter · Decision Letter 1]

18 Jan 2021

PONE-D-20-13979R1

HbA1c is inversely associated with thyroid cysts in a euthyroid population: a cross-sectional study

PLOS ONE

Dear Dr. Shimizu,

Thank you for submitting your manuscript to PLOS ONE. After careful consideration, we feel that it has merit but does not fully meet PLOS ONE’s publication criteria as it currently stands. Therefore, we invite you to submit a revised version of the manuscript that addresses the points raised during the review process.

We look forward to receiving your revised manuscript.

Kind regards,

Yoshihiro Kokubo, PhD, MD, FACC, FAHA, FESC, FESO

Academic Editor

PLOS ONE

Additional Editor Comments (if provided):

Thank you for correcting the treatise.

There is a contradiction between the results and considerations that the authors have derived.

I think that one of the causes of the contradiction is due to cross-sectional research. The authors should carefully discuss this in this manuscript.

Secondly, since the cut-off value of the third quantile of HbA1c is within the normal range, it would be meaningless to classify within that. Therefore, the cut-off value of HbA1c should be the value used in clinical diagnosis. The peer reviewers were not interested in the numerous peer review requests. It will be necessary to devise ways to get the reader interested.

One reviewer's comments are very reasonable. Please reconsider faithfully to this comment.

Reviewers' comments:

Reviewer's Responses to Questions

**Comments to the Author**

1. If the authors have adequately addressed your comments raised in a previous round of review and you feel that this manuscript is now acceptable for publication, you may indicate that here to bypass the “Comments to the Author” section, enter your conflict of interest statement in the “Confidential to Editor” section, and submit your "Accept" recommendation.

Reviewer #2: (No Response)

Reviewer #3: All comments have been addressed

2. Is the manuscript technically sound, and do the data support the conclusions?

Reviewer #2: Partly

Reviewer #3: Yes

3. Has the statistical analysis been performed appropriately and rigorously? 

Reviewer #2: No

Reviewer #3: Yes

4. Have the authors made all data underlying the findings in their manuscript fully available?

Reviewer #2: No

Reviewer #3: Yes

5. Is the manuscript presented in an intelligible fashion and written in standard English?

Reviewer #2: Yes

Reviewer #3: Yes

6. Review Comments to the Author

Reviewer #2: I think your conclusion is unreasonable from your results. First, in lines 50-53 in the Abstract, you wrote “Previously, we reported that euthyroid individuals without thyroid cysts had a stronger correlation between TSH and free T3 than those with thyroid cysts [3]. Therefore, the absence of thyroid cysts might negatively affect thyroid hormone synthesis.” However, those correlations were r=-0.13 (p<0.001) in Thyroid cyst (-) group and r=-0.03 (p=0.525) in Thyroid cyst (+) group. In lines 232-237 in the Discussion, you wrote “As the absence of thyroid cysts might indicate latent damage to the thyroid [3,4,7], insulin resistance related to TSH levels might underlie the association between thyroid cysts and HbA1c. In fact, in our additional analysis, a slight but significant positive association between TSH and HbA1c levels was observed, while significant inverse associations between TSH and thyroid hormone (free T3 and free T4) levels were observed among all euthyroid participants (Table 3).” However, those results showed r<0.15 (absolute value). Your interpretation of those results does not seem appropriate. If a Correlation Coefficient is less than 0.2 (absolute value), it is considered negligible. Even if it has statistical significance, it doesn’t necessarily mean it has medical significance. The odds ratio in Table 5 for Models 1, 2, 3 are around 0.75. Those don’t seem to indicate strong associations. Based on the above points, I think you are jumping to the conclusion that your findings provide an efficient tool to evaluate the thyroid hormone activity that relates to insulin resistance.

Furthermore, in line 119, you wrote “… to determine the association between TPO-Ab and thyroid cysts”, but you didn’t measure TPO-Ab. Therefore, you cannot determine that association. That kind of statement is confusing for the reader.

Reviewer #3: Revisions are extremely helpful. The English is now clear and the import of the report is evident.

7. PLOS authors have the option to publish the peer review history of their article (what does this mean?). If published, this will include your full peer review and any attached files.

Reviewer #2: **Yes: **Mihoko Takahashi, PhD

Reviewer #3: **Yes: **Donald Zimmerman

---

## [Author Response · Author response to Decision Letter 1]

22 Jan 2021

Reviewer #2:

1) 

I think your conclusion is unreasonable from your results. First, in lines 50-53 in the Abstract, you wrote “Previously, we reported that euthyroid individuals without thyroid cysts had a stronger correlation between TSH and free T3 than those with thyroid cysts [3]. Therefore, the absence of thyroid cysts might negatively affect thyroid hormone synthesis.” However, those correlations were r=-0.13 (p<0.001) in Thyroid cyst (-) group and r=-0.03 (p=0.525) in Thyroid cyst (+) group. In lines 232-237 in the Discussion, you wrote “As the absence of thyroid cysts might indicate latent damage to the thyroid [3,4,7], insulin resistance related to TSH levels might underlie the association between thyroid cysts and HbA1c. In fact, in our additional analysis, a slight but significant positive association between TSH and HbA1c levels was observed, while significant inverse associations between TSH and thyroid hormone (free T3 and free T4) levels were observed among all euthyroid participants (Table 3).” However, those results showed r<0.15 (absolute value). Your interpretation of those results does not seem appropriate. If a Correlation Coefficient is less than 0.2 (absolute value), it is considered negligible. Even if it has statistical significance, it doesn’t necessarily mean it has medical significance. 

→

Thank you for valuable comment. According to this valuable comment, I rechecked the present conclusions and potential mechanism that might underlying present results. 

I’m completely agreed with this reviewer’s comment that showed even the statistical power showed the significant value, correlation coefficient less than 0.2 is clinically meaning less. 

Therefore, as we described in discussion section that we thought the sensitivity of using the absence of thyroid cysts and higher levels of HbA1c to detect latent damage to thyroid could be much greater than that of using TSH. 

To clarify those associations, we added the analysis that clarified the association between HbA1c and subclinical hypothyroidism because latent damage of thyroid might be associated with hypothyroidism. 

And we found significant positive association between HbA1c and subclinical hypothyroidism. Therefore, we thought, HbA1c could indicate latent damage of thyroid. 

Then we added following sentences in 

Odds ratios (ORs) and 95% confidence intervals (CIs) were calculated using logistic regression models to assess association between HbA1c and subclinical hypothyroidism.

And we also added following sentences in result section with table.

Association between subclinical hypothyroidism and HbA1c

 Table 3 shows a significant positive association between HbA1c and subclinical hypothyroidism that is independent of known cardiovascular risk factors.

We also added following sentences in discussion section.

In the present study, we found a significant positive association between subclinical hypothyroidism and HbA1c (Table 3). This is consistent with the findings of a previous study that reported the levels of HbA1c in non-diabetic participants with subclinical hypothyroidism to be higher than those in normal healthy controls [7].

However, HbA1c levels is not correlated with thyroid hormone (free T3 and free T4). And even HbA1c shows slightly significantly positively correlated with TSH, the magnitude of this correlation is too small to explain present results. Therefore, we thought the sensitivity of HbA1c that detecting latent damage of thyroid might be stronger than TSH.

Previous study reports that lower thyroid function may lead to extended longevity among elderly [Ref1]. The we thought the demands of thyroid decreased with process of aging. Therefore, not only actual serum concentration of thyroid hormone levels but also demands of thyroid hormone might determine the secretion levels of TSH. However, equilibrium between thyroid hormone and glucose might determine the insulin resistance [Ref2], HbA1c could indicate the deficiency of thyroid hormone.

[Ref1]

Gesing A, Lewiński A, Karbownik-Lewińska M. The thyroid gland and the process of aging; what is new? Thyroid Res. 2012;5(1):16. doi: 10.1186/1756-6614-5-16.

[Ref2]

Brenta G. Why can insulin resistance be a natural consequence of thyroid dysfunction? J Thyroid Res. 2011;2011:152850.

Then we added following sentences that explains potential mechanism of present results in discussion section.

The magnitude of the observed associations (between TSH and HbA1c, and between TSH and thyroid hormones) were too small to explain the present results showing a significant inverse association between HbA1c and thyroid cysts. The magnitude of physiological demand for thyroid hormone activity might explain this discrepancy. Because decreased thyroid function may extend longevity, the demand for thyroid hormone activity may decrease with age [16]. Therefore, not only the serum thyroid hormone level, but also the demand for thyroid activity might determine the level of TSH secretion. HbA1c levels may indicate the magnitude of thyroid hormone deficiency in the same way that the equilibrium between thyroid hormone and glucose might determine insulin resistance [15]. In the present study, HbA1c was significantly positively correlated with subclinical hypothyroidism (Table 3) but not with thyroid hormone (Table 4). Because the absence of thyroid hormone indicates latent damage to the thyroid [3,5,8], HbA1c levels could inversely associate with thyroid cysts (Table 5). 

The same associations between HbA1c levels and thyroid cysts were observed when analyses were limited to the normal TSH range (Table 5). 

Therefore, the sensitivity of using the absence of thyroid cysts and higher levels of HbA1c to detect latent damage to the thyroid could be much greater than that of using TSH level.

And we deleted the description of the correlation among thyroid cysts, thyroid hormone and TSH in abstract section and introduction section.

Furthermore, we added the following sentence in introduction section that partly indicates that subjects with thyroid cyst could possess higher activity of thyroid hormone.

We previously identified a significant positive association between thyroid cysts and isolated systolic hypertension [3]. Since hyperthyroidism is known to be a common cause of isolated systolic hypertension [4], development of thyroid cysts could benefit patients with low thyroid hormone synthesis, and absence of thyroid cysts might attenuate thyroid hormone synthesis.

And we added the definition of subclinical hypothyroidism as following.

 The threshold for subclinical hypothyroidism is set at TSH > 4.01 μIU/mL.

2)

The odds ratio in Table 5 for Models 1, 2, 3 are around 0.75. Those don’t seem to indicate strong associations. Based on the above points, I think you are jumping to the conclusion that your findings provide an efficient tool to evaluate the thyroid hormone activity that relates to insulin resistance.

→

Thank you for valuable comment. Since most of those subjects showed normal range of HbA1c, we revised to use the clinical cut off values of HbA1c. And we found that compared to the subjects with completely normal range of HbA1c (<5.6%), subjects with range of diabetes (6.5% ≤ HbA1c) shows 0.48 (0.28, 0.81). When we excluded the subjects with taking glucose lowering medication, the value became 0.38 (0.16, 0.88). Therefore, HbA1c might strongly inversely associated with thyroid cysts.

3) 

Furthermore, in line 119, you wrote “… to determine the association between TPO-Ab and thyroid cysts”, but you didn’t measure TPO-Ab. Therefore, you cannot determine that association. That kind of statement is confusing for the reader.

→

This is completely typo. Sorry for inconvenience. I corrected. 

Reviewer #3: Revisions are extremely helpful. The English is now clear and the import of the report is evident.

→

Thank you for valuable comment.

---

## [Decision Letter · Decision Letter 2]

18 Mar 2021

PONE-D-20-13979R2

HbA1c is inversely associated with thyroid cysts in a euthyroid population: a cross-sectional study

PLOS ONE

Dear Dr. Shimizu,

Thank you for submitting your manuscript to PLOS ONE. After careful consideration, we feel that it has merit but does not fully meet PLOS ONE’s publication criteria as it currently stands. Therefore, we invite you to submit a revised version of the manuscript that addresses the points raised during the review process.

In particular, I recommed to change the conclusions of your manuscript according to the indications of Reviewer  2.

We look forward to receiving your revised manuscript.

Kind regards,

Silvia Naitza

Academic Editor

PLOS ONE

Additional Editor Comments (if provided):

Dear Dr. Shimizu,

please find attached the comments from the Reviewers to the revised version 2 of your manuscript PONE-D-20-13979R2. As you can see, one Reviewer has still major concerns especially with the conclusions of your manuscript, which I also share. Please, change your manuscript accordingly before resubmitting a new version addressing the reviewers criticisms.

Best regards,

Silvia Naitza

Reviewers' comments:

Reviewer's Responses to Questions

**Comments to the Author**

1. If the authors have adequately addressed your comments raised in a previous round of review and you feel that this manuscript is now acceptable for publication, you may indicate that here to bypass the “Comments to the Author” section, enter your conflict of interest statement in the “Confidential to Editor” section, and submit your "Accept" recommendation.

Reviewer #2: All comments have been addressed

Reviewer #3: (No Response)

2. Is the manuscript technically sound, and do the data support the conclusions?

Reviewer #2: Partly

Reviewer #3: (No Response)

3. Has the statistical analysis been performed appropriately and rigorously? 

Reviewer #2: Yes

Reviewer #3: (No Response)

4. Have the authors made all data underlying the findings in their manuscript fully available?

Reviewer #2: No

Reviewer #3: (No Response)

5. Is the manuscript presented in an intelligible fashion and written in standard English?

Reviewer #2: Yes

Reviewer #3: (No Response)

6. Review Comments to the Author

Reviewer #2: Thank you for the revision. The results were interesting. It might be valuable for publication. However, I still don’t agree with your conclusion.

In the discussion, you wrote “the absence of thyroid cysts might indicate latent damage to the thyroid" (line 223) and “which could not be evaluated based on TSH, free T3 and free T4 levels" (lines 269-270). It sounds like you are suggesting a hypothesis that the absence of thyroid cysts could be used to diagnose latent damage and thyroid activity.

I could understand if you said the existence of thyroid cysts might indicate a high level of thyroid activity. However, it is difficult to say the absence of thyroid cysts indicate latent damage. Among the people without thyroid cysts, there must be a lot of people with completely healthy thyroids. Isn’t it more appropriate to think there is only a small fraction of people with latent damage? Jumping to conclusions like that will lead to mistaken diagnoses.

Reviewer #3: 1. Abstract first sentence-- Change to: Previously, anti-peroxidase antibodies (TPO-Ab), which are known to cause autoimmune thyroiditis, were shown to be present in individuals in inverse proportion to the prevalence of thyroid cysts.

2. Abstract third sentence--Change to: Since participants with subclinical hypothyroidism are reported to have higher HbA1c than do normal healthy controls, HbA1c could be inversely associated with the incidence of thyroid cysts through a mechanism reflecting latent thyroid damage.

7. PLOS authors have the option to publish the peer review history of their article (what does this mean?). If published, this will include your full peer review and any attached files.

Reviewer #2: **Yes: **Mihoko Takahashi, PhD

Reviewer #3: **Yes: **Donald Zimmerman,MD

---

## [Author Response · Author response to Decision Letter 2]

21 Mar 2021

Reviewer #2: Thank you for the revision. The results were interesting. It might be valuable for publication. However, I still don’t agree with your conclusion.

In the discussion, you wrote “the absence of thyroid cysts might indicate latent damage to the thyroid" (line 223) and “which could not be evaluated based on TSH, free T3 and free T4 levels" (lines 269-270). It sounds like you are suggesting a hypothesis that the absence of thyroid cysts could be used to diagnose latent damage and thyroid activity.

I could understand if you said the existence of thyroid cysts might indicate a high level of thyroid activity. However, it is difficult to say the absence of thyroid cysts indicate latent damage. Among the people without thyroid cysts, there must be a lot of people with completely healthy thyroids. Isn’t it more appropriate to think there is only a small fraction of people with latent damage? Jumping to conclusions like that will lead to mistaken diagnoses.

➡

(Reply)

Thank you for your valuable comment. According to this valuable comment, we reconsider the scientific meaning of present results.

As shown in Table 3, we found significant positive association between HbA1c and subclinical hypothyroidisms. This indicates that, high level of HbA1c is associated with subclinical thyroid disorder. 

And as shown in Table 5, we found significant inverse association between HbA1c and thyroid cysts. 

Then in present study, high level of HbA1c that could be associated with subclinical thyroid disorder revealed to be inversely associated with thyroid cysts. This is why we thought absence of thyroid cyst might indicate latent thyroid damage that is associated with high levels of HbA1c (insulin resistance). 

In order to show the scientific evidence that supports “existence of thyroid cysts might indicate a high levels of activity” as this reviewer commented, we have to perform other analyzes that showed two associations. The first one is an inverse association between HbA1c and subclinical hyperthyroidism. And second one is a positive association between HbA1c and thyroid cysts.

Even HbA1c shows positively associated with subclinical hypothyroidism as shown in Table 2, this fact never indicate that HbA1c is inversely associated with subclinical hyperthyroidism. In order to describe about the influence of higher activity of thyroid hormone, we should show significant inverse association between subclinical hyperthyroidism and HbA1c.

In addition to that, even hypothyroidism is reported to be associated with diabetes [Ref1], another study also reports that not only serum TSH levels but also T3 levels were increased in participants with diabetes [Ref2]. This means both of hyperthyroidism and hypothyroidism might increase the level of HbA1c. Therefore, positive association between HbA1c and thyroid cysts should be observed if high levels of thyroid hormone activity that associated with thyroid cysts causes main results of present study. However, we found significant inverse association between HbA1c and thyroid cysts.

[Ref1]

Makada MG et al. Study of glycated haemoglobin (HbA1c) in non-diabetic subjects with subclinical hypothyroidism. J Clin Diagn Res. 2017;11(4):BC01-04.

[Ref2]

Elgazar FH, et al. Thyroid dysfunction prevalence and relation to glycemic control in patients with type 2 diabetes mellitus. Diabetes Metab Syndr. 2019;13(4):2513-2517.

This is why we can’t describe as existence of thyroid cysts might reduce the HbA1c in present study by indicating a high level activity of thyroid hormone. Then we did describes as following in discussion section.

Both hyperthyroidism and hypothyroidism are reported to be associated with the development of insulin resistance [12]. However, a positive linear association between the normal range of TSH level and insulin resistance among non-diabetes participants and type 2 diabetes patients has been reported [13]. Furthermore, a positive correlation between HbA1c and TSH levels was also observed in patients with subclinical hypothyroidism [14].

Even though, I also could understand the meaning that this reviewer thought that there are too many completely healthy participants without thyroid cysts. 

However, we also have to regard the fact that we have no efficient diagnostic tool to evaluate the latent thyroid damage that could not be evaluated by thyroid hormones and thyroid stimulating hormone. 

Then we could not figure out the exact prevalence rate of participants with completely healthy thyroid. In fact, there is growing evidence suggesting that the treatment of subclinical hypothyroidism may not be beneficial, particularly in older patients [Ref3]. However, aging patients may progress from subclinical hypothyroidism to overt hypothyroidism. Therefore, determining the timing to begin treatment for hypothyroidism is difficult in daily clinical practice [Ref4]. This difficulty might partly be caused by latent damage of thyroid and decreased demand of thyroid activity along with process of aging. 

[Ref3]

Leng O, Razvi S. Hypothyroidism in the older population. Thyroid Res. 2019;12.2 doi: 10.1186/s13044-019-0063-3.

[Ref4]

Khandelwal D, Tandon N. Overt and subclinical hypothyroidism: who to treat and how. Drugs. 2012;72(1):17-33.

This is why we described the sentence “at least partly” as following in abstract.

Therefore, at least partly, absence of thyroid cysts could indicate latent thyroid damage.

And we did describe the potential function that thyroid cyst might possesses as following in introduction section.

We previously identified a significant positive association between thyroid cysts and isolated systolic hypertension [3]. Since hyperthyroidism is known to be a common cause of isolated systolic hypertension [4], development of thyroid cysts could benefit patients with low thyroid hormone synthesis, and absence of thyroid cysts might attenuate thyroid hormone synthesis.

This description is also important because this sentences indicates that thyroid cysts can support thyroid hormone synthesis but not activates aggressive thyroid hormone synthesis. And we described why absence of thyroid cysts could be associated with latent thyroid damage as following in introduction section.

As TPO-Ab inhibits thyroid peroxidase, which plays an important role in synthesizing thyroid hormones [6], the absence of thyroid cysts in euthyroid individuals could indicate latent damage to the thyroid.

Even this reviewer commented as “It sound like you are suggesting a hypothesis that the absence of thyroid cysts could be used to diagnose latent damage and thyroid activity.”, we never thought that the presence of latent thyroid damage could be estimated only by absence of thyroid cysts. This is why we did describe as following in discussion section. 

the sensitivity of using the absence of thyroid cysts and higher levels of HbA1c to detect latent damage to the thyroid could be much greater than that of using TSH level.

And because our present study is not intended to describe that HbA1c is efficient tool to evaluate the latent damage of thyroid but to describe that status of thyroid cysts could be influence on hormone activity that relates to insulin resistance. Then we describe in conclusion as “Although further investigation on the present topic are necessary, the present finding provides an efficient tool to evaluate the thyroid hormone activity relative to insulin resistance among participants with normal thyroid function.

 Because of those above mentioned reasons, I’m disagree with this reviewer’s valuable comment. 

#3 Reviewer: Previously, anti-peroxidase antibodies (TPO-Ab), which are known cause autoimmune thyroiditis, were shown to be present in individuals in inverse proportion to the prevalence of thyroid cysts.

➡(Reply)

Thank you for valuable comment. According to this reviewer’s valuable comment, we re-checked the content of present study. Since we evaluated the existence of a thyroid cysts on the parameter whether it was present or not, we did not evaluate the association between the number of thyroid cysts (size of the thyroid cysts) and HbA1c. This is important fact that should be taken into consideration. Therefore, we thought the sentence “were shown to be present in individuals in inverse proportion to the prevalence of thyroid cysts” might not appropriate to use. Then we described as following. 

Previously, anti-peroxidase antibody (TPO-Ab) which is known cause of autoimmune thyroiditis is revealed to be inversely associated with thyroid cysts among euthyroid population. 

➡(Reply)

Thank you for valuable comment. According to this reviewer’s valuable comment, we re-checked the content of present study. Since present study is a cross-sectional study, we could not determine the causal relationships. Therefore, the sentence “incidence of” is not appropriate to use. Then we revised as following. 

Since participants with subclinical hypothyroidism are reported to have higher HbA1c than normal healthy controls, HbA1c could be inversely associated with presence of thyroid cysts through a mechanism reflecting latent thyroid damage.

---

## [Decision Letter · Decision Letter 3]

28 Apr 2021

PONE-D-20-13979R3

HbA1c is inversely associated with thyroid cysts in a euthyroid population: a cross-sectional study

PLOS ONE

Dear Dr. Shimizu,

Thank you for submitting your manuscript to PLOS ONE. After careful consideration, we feel that it has merit but does not fully meet PLOS ONE’s publication criteria as it currently stands. Therefore, we invite you to submit a revised version of the manuscript that addresses the points raised during the review process.

In particular, please modify the sentences highlighted by reviewer 2 to improve clarity and refrase some of the paragraphs in your conclusions in a more dubitative form, leaving an hypothesis open, as suggested by reviewer 3.

We look forward to receiving your revised manuscript.

Kind regards,

Silvia Naitza

Academic Editor

PLOS ONE

Additional Editor Comments (if provided):

Dear Dr. Shimizu,

please find enclosed the comments of the three referees that have now evaluated your revised manuscript PONE-D-20-13979R3. Although the manuscript has its value and has been improved following the referees' advice, there is still concern about its suitability for publication in Plos ONE as it stands now. In particular, the language can be further improved as the meaning of some sentences is not very clear (see reviewer 2) and the interpretation of the results is rather provocative, which led one of the reviewer to suggest rejection of your manuscript. I invite you to address the requests of these reviewers and refrase some of the paragraphs in your conclusions in a more dubitative form, leaving an hypothesis open before resubmitting a revised version to the journal.

Best regards,

Silvia Naitza

Reviewers' comments:

Reviewer's Responses to Questions

**Comments to the Author**

1. If the authors have adequately addressed your comments raised in a previous round of review and you feel that this manuscript is now acceptable for publication, you may indicate that here to bypass the “Comments to the Author” section, enter your conflict of interest statement in the “Confidential to Editor” section, and submit your "Accept" recommendation.

Reviewer #1: All comments have been addressed

Reviewer #2: (No Response)

Reviewer #3: All comments have been addressed

2. Is the manuscript technically sound, and do the data support the conclusions?

Reviewer #1: Partly

Reviewer #2: No

Reviewer #3: Yes

3. Has the statistical analysis been performed appropriately and rigorously? 

Reviewer #1: Yes

Reviewer #2: Yes

Reviewer #3: Yes

4. Have the authors made all data underlying the findings in their manuscript fully available?

Reviewer #1: Yes

Reviewer #2: No

Reviewer #3: Yes

5. Is the manuscript presented in an intelligible fashion and written in standard English?

Reviewer #1: No

Reviewer #2: Yes

Reviewer #3: Yes

6. Review Comments to the Author

Reviewer #1: Even though the manuscript has been improved there are still some concerns with the meaning of some sentences that would need to be modified.

1)Line 20

“Previously, anti-peroxidase antibody (TPO-Ab) which is known cause of autoimmune

thyroiditis is revealed to be inversely…”

I would suggest rephrasing the following sentences for clarity.

2) Line 38

“investigations are necessary, the present findings help efficiently evaluate thyroid

hormone activity corresponding to insulin resistance among participants with normal…”

Please rephrase since the meaning of the sentence is not clear

3) Line 49

“and isolated systolic hypertension [3]. Since hyperthyroidism is known to be a common

cause of isolated systolic hypertension [4], development of thyroid cysts could benefit

patients with low thyroid hormone synthesis, and absence of thyroid cysts might

attenuate thyroid hormone synthesis.”

I understand the Authors want to state that the absence of thyroid cysts might reflect attenuated thyroid hormone synthesis but the whole sentence does not make sense . Please rephrase.

4) Line 110

6.5≤ Does it means less than 6.5?

5)Footnote table 3: please correct “HDLc”

6) Title table 6: please correct “without taking glucose lowering”

7)Line 255: “Therefore, the sensitivity of using the absence of thyroid cysts and higher levels of

HbA1c to detect latent damage to the thyroid could be much greater than that of using

TSH level.”

I would suggest changing this sentence because the sensitivity of these putative diagnostic tools has not been assessed in this study and cannot replace a standardized biochemical test such as TSH that has been traditionally used to detect hypothyroidism.

Reviewer #2: Thank you for the reply.

I read this manuscript again after reading your explanation. However, my concern still remains. In fact, you wrote “I also could understand the meaning that this reviewer thought that there are too many completely healthy participants without thyroid cysts.” However, you didn’t discuss that in the manuscript. Thus, you didn’t reflect my comments at all, even though you agree with part of my comments, nor did you provide a satisfactory explanation for the part of my comments with which you disagreed. Consequently, I recommend rejecting this manuscript.

Reviewer #3: The manuscript reads well and is of considerable interest.

The relation between thyroid cysts and hemoglobin A1C is well established.

7. PLOS authors have the option to publish the peer review history of their article (what does this mean?). If published, this will include your full peer review and any attached files.

Reviewer #1: **Yes: **Gabriela Brenta

Reviewer #2: **Yes: **Mikoho Takahashi, PhD

Reviewer #3: **Yes: **Donald Zimmerman

---

## [Author Response · Author response to Decision Letter 3]

29 Apr 2021

Reviewer #1: Even though the manuscript has been improved there are still some concerns with the meaning of some sentences that would need to be modified.

1)Line 20

“Previously, anti-peroxidase antibody (TPO-Ab) which is known cause of autoimmune

thyroiditis is revealed to be inversely…”

I would suggest rephrasing the following sentences for clarity.

→

Thank you for your valuable comment. According to this valuable comment, we revised as following.

 Anti-peroxidase antibody (TPO-Ab) is revealed to be inversely associated with thyroid cysts among euthyroid population. TPO-Ab causes autoimmune thyroiditis by bolstering thyroid inflammation. 

2) Line 38

“investigations are necessary, the present findings help efficiently evaluate thyroid

hormone activity corresponding to insulin resistance among participants with normal…”

Please rephrase since the meaning of the sentence is not clear

→

Thank you for your valuable comment. According to this valuable comment, we reconsider the meaning of this sentence. Since the fact that further investigation is necessary on present topic are the things to taken for granted, we delete the sentence. And simplify as following.

 Present finding provides an efficient tool to evaluate the thyroid hormone activity relative to insulin resistance among participants with normal thyroid function. 

3) Line 49

“and isolated systolic hypertension [3]. Since hyperthyroidism is known to be a common

cause of isolated systolic hypertension [4], development of thyroid cysts could benefit

patients with low thyroid hormone synthesis, and absence of thyroid cysts might

attenuate thyroid hormone synthesis.”

I understand the Authors want to state that the absence of thyroid cysts might reflect attenuated thyroid hormone synthesis but the whole sentence does not make sense . Please rephrase.

→

Thank you for your valuable comment. According to this valuable comment, we reconsider the meaning of this sentence. Hyperthyroidism is known major secondary causes of isolated systolic hypertension. And because we found significant positive association between thyroid cysts and isolated hypertension in our previous study with euthyroid participants, we could described that euthyriod participants with thyroid cyst might have relatively higher thyroid hormone synthesizing activity than those without thyroid cysts. But the sentence that “the absence of thyroid cysts might reflect attenuated thyroid hormone synthesis” is not appropriate. Since we described that the absence of thyroid cysts in euthyroid individuals could indicate latent damage to the thyroid in next section, we revised as following. 

 We previously identified a significant positive association between thyroid cysts and isolated systolic hypertension among euthyroid participants [3]. Since hyperthyroidism is a major secondary cause of isolated hypertension [4], euthyroid participants with thyroid cysts might have relatively higher thyroid hormone synthesizing activity than those without thyroid cysts. 

4) Line 110

6.5≤ Does it means less than 6.5?

→

Thank you for your valuable comment. According to this valuable comment, we recheck the relevant sentence. This is HbA1c level (<5.6% for [low], 5.6-6.4 % for [medium], and 6.5% ≤ for [high]). Then meaning of 6.5% ≤ was Hba1c levels as and above 6.5%. In tabled we also described as <5.6%, 5.6-6.4 % , and 6.5% ≤ ). Therefore, <5.6% means less than 5.6%. To avoid misunderstand, we added the word [low] [medium] and [high] for tables. 

5)Footnote table 3: please correct “HDLc”

→

Thank you for your valuable comment. According to this valuable comment, we recheck the relevant sentence. Sorry for inconvenience, I corrected. 

6) Title table 6: please correct “without taking glucose lowering”

→

Thank you for your valuable comment. According to this valuable comment, we recheck the relevant sentence. Since we used as “not receiving glucose-lowering medication” we revised as so. 

7)Line 255: “Therefore, the sensitivity of using the absence of thyroid cysts and higher levels of

HbA1c to detect latent damage to the thyroid could be much greater than that of using

TSH level.”

I would suggest changing this sentence because the sensitivity of these putative diagnostic tools has not been assessed in this study and cannot replace a standardized biochemical test such as TSH that has been traditionally used to detect hypothyroidism.

→

Thank you for your valuable comment. According to this valuable comment, we reconsider the relevant sentence. And I revised as following, 

Therefore, the absence of thyroid cysts and higher levels of HbA1c could indicate the latent functional damage of the thyroid.

Reviewer #2: Thank you for the reply.

I read this manuscript again after reading your explanation. However, my concern still remains. In fact, you wrote “I also could understand the meaning that this reviewer thought that there are too many completely healthy participants without thyroid cysts.” However, you didn’t discuss that in the manuscript. Thus, you didn’t reflect my comments at all, even though you agree with part of my comments, nor did you provide a satisfactory explanation for the part of my comments with which you disagreed. Consequently, I recommend rejecting this manuscript.

→

Thank you for your valuable comment. Even this reviewer thought that there is no description about the completely healthy participants without thyroid cysts, we already described those sentences with closed meaning as I made reply before. Those sentences describe that the thyroid cysts might have the clinical feature on thyroid function and absence of thyroid hormone might have disadvantage on synthesizing thyroid hormone with same references. 

In addition to that, we also described about the age-related reduction of demand of thyroid hormone as following.

However, the magnitude of the observed associations (between TSH and HbA1c, and between TSH and thyroid hormones) were too small to explain the present results showing a significant inverse association between HbA1c and thyroid cysts. The magnitude of physiological demand for thyroid hormone activity might explain this discrepancy. Because decreased thyroid function may extend longevity, the demand for thyroid hormone activity may decrease with age [16]. Therefore, not only the serum thyroid hormone level, but also the demand for thyroid activity might determine the level of TSH secretion.

Those description indicates that, diagnosing as completely healthy thyroid condition only by thyroid hormone level which is not considering about age-related demand level is inappropriate. And making criteria to diagnose completely healthy thyroid is not main topic of present study. Then we reconsider what sentence could be added without deteriorating scientific meaning to this review’s comment. To diagnose as having completely healthy thyroid condition among euthyroid individuals, at least we should evaluate the turnover of thyroid hormone which is not common for annual health check-up and daily clinical practice but needs the technique of nuclear medicine. Since we could not evaluate the completely healthy condition and the word “completely healthy condition” is not scientific, we added following sentences in limitation section. 

Third, the turnover of the thyroid hormone may be influenced by status of thyroid cysts, but we could not evaluate the half-life of thyroid hormone. 

Reviewer #3: The manuscript reads well and is of considerable interest.

→

Thank you for your valuable comment.

---

## [Decision Letter · Decision Letter 4]

2 Jun 2021

PONE-D-20-13979R4

HbA1c is inversely associated with thyroid cysts in a euthyroid population: a cross-sectional study

PLOS ONE

Dear Dr. Shimizu,

Thank you for submitting your manuscript to PLOS ONE. After careful consideration, we feel that it has merit but does not fully meet PLOS ONE’s publication criteria as it currently stands. Therefore, we invite you to submit a revised version of the manuscript that addresses the points raised during the review process.

In particular, you should modify your manuscript in the points suggested by reviewer 2 and then resubmit to the Jounal.

We look forward to receiving your revised manuscript.

Kind regards,

Silvia Naitza

Academic Editor

PLOS ONE

Journal Requirements:

Reviewers' comments:

Reviewer's Responses to Questions

**Comments to the Author**

1. If the authors have adequately addressed your comments raised in a previous round of review and you feel that this manuscript is now acceptable for publication, you may indicate that here to bypass the “Comments to the Author” section, enter your conflict of interest statement in the “Confidential to Editor” section, and submit your "Accept" recommendation.

Reviewer #1: All comments have been addressed

Reviewer #2: (No Response)

2. Is the manuscript technically sound, and do the data support the conclusions?

Reviewer #1: Yes

Reviewer #2: No

3. Has the statistical analysis been performed appropriately and rigorously? 

Reviewer #1: Yes

Reviewer #2: Yes

4. Have the authors made all data underlying the findings in their manuscript fully available?

Reviewer #1: Yes

Reviewer #2: Yes

5. Is the manuscript presented in an intelligible fashion and written in standard English?

Reviewer #1: Yes

Reviewer #2: Yes

6. Review Comments to the Author

Reviewer #1: All comments have been adequately answered

I am still in doubt if the p values in table 6 are correct since they are non significant.

Reviewer #2: Thank you for the revision. Since you deleted the sentence, “Therefore, the sensitivity of using the absence of thyroid cysts and higher levels of HbA1c to detect latent damage to the thyroid could be much greater than that of using TSH level.”, I am able to accept your logic a little. However, I still disagree with your conclusion. In the conclusion, you wrote, “the present finding provides an efficient tool to evaluate the thyroid hormone activity relative to insulin resistance among participants with normal thyroid function.” However, this conclusion was based on your hypothesis, not your results. You should put, “The absence of thyroid cysts and higher levels of HbA1c could indicate the latent functional damage of the thyroid” into the Conclusion, as you wrote in the Discussion, and delete “the present finding provides an efficient tool to evaluate the thyroid hormone activity relative to insulin resistance among participants with normal thyroid function.” The same is true about the Abstract. Furthermore, in lines 274-275 in the Discussion, you wrote, “the present study also provides an efficient tool to evaluate the validity of HbA1c level control in participants with normal thyroid function.” I think you should also change this sentence for the same reason.

7. PLOS authors have the option to publish the peer review history of their article (what does this mean?). If published, this will include your full peer review and any attached files.

Reviewer #1: **Yes: **Gabriela Brenta

Reviewer #2: **Yes: **Mihoko Takahashi

---

## [Author Response · Author response to Decision Letter 4]

2 Jun 2021

Reviewer #1: All comments have been adequately answered

I am still in doubt if the p values in table 6 are correct since they are non significant.

→

Thank you for your valuable comment. In present study, we categorized HbA1c as <5.6%, 5.6-6.4%, and 6.5%≤. The analysis that is shown in Table 6 is limited to participants without taking glucose lowering medication. Then the number of participants with normal range of HbA1c is much higher than that of high HbA1c level. Among not taking glucose lowering medication, the number of participant (case of thyroid cysts) in each category are 983 (319), 584 (205) and 52 (10). Then those categories are biased in number. That results in p>0.05 level when considering trends as shown in table 6. Even thought, when we made those analyzes by using tertile categories of HbA1c, the number of participants and thyroid cases were 543 (176) for T1, 530 (188) for T2 and 434 (136) for T3. And the p-values were 0.021 for Model 1, 0.025 for Model 2 and 0.0446 for Model 3. Therefore, the findings that this reviewer pointed out is the matter of distribution. Possibly, the wide range of 95% CI and Odds shows nearly 1 for subjects with HbA1c (5.6-6.4%), might results in no significant values when considering p as trend. 

Reviewer #2: Thank you for the revision. Since you deleted the sentence, “Therefore, the sensitivity of using the absence of thyroid cysts and higher levels of HbA1c to detect latent damage to the thyroid could be much greater than that of using TSH level.”, I am able to accept your logic a little. However, I still disagree with your conclusion. In the conclusion, you wrote, “the present finding provides an efficient tool to evaluate the thyroid hormone activity relative to insulin resistance among participants with normal thyroid function.” However, this conclusion was based on your hypothesis, not your results.

 You should put, “The absence of thyroid cysts and higher levels of HbA1c could indicate the latent functional damage of the thyroid” into the Conclusion, as you wrote in the Discussion, and delete “the present finding provides an efficient tool to evaluate the thyroid hormone activity relative to insulin resistance among participants with normal thyroid function.” The same is true about the Abstract. 

→

Thank you for your valuable comment. According to this valuable comment, I deleted the sentence “the present finding provides an efficient tool to evaluate the thyroid hormone activity relative to insulin resistance among participants with normal thyroid function” from conclusion section of abstract and discussion. And described as “The absence of thyroid cysts and higher levels of HbA1c could indicate the latent functional damage of the thyroid”.

Furthermore, in lines 274-275 in the Discussion, you wrote, “the present study also provides an efficient tool to evaluate the validity of HbA1c level control in participants with normal thyroid function.” I think you should also change this sentence for the same reason.

→

Thank you for your valuable comment. According to this valuable comment, I deleted the sentence “he present study also provides an efficient tool to evaluate the validity of HbA1c level control in participants with normal thyroid function.” Since the last sentence indicates the importance of Hba1c level control in subjects with subclinical hypothyroid, we added the sentence “Present findings could provide efficient cues to control HbA1c associated with the latent functional damage of the thyroid.”

---

## [Editor Report · Decision Letter 5]

15 Jun 2021

HbA1c is inversely associated with thyroid cysts in a euthyroid population: a cross-sectional study

PONE-D-20-13979R5

Dear Dr. Shimizu,

We’re pleased to inform you that your manuscript has been judged scientifically suitable for publication and will be formally accepted for publication once it meets all outstanding technical requirements.

Kind regards,

Silvia Naitza

Academic Editor

PLOS ONE
---

## [Editor Report · Acceptance letter]

22 Jun 2021

PONE-D-20-13979R5 

HbA1c is inversely associated with thyroid cysts in a euthyroid population: a cross-sectional study 

Dear Dr. Shimizu:

I'm pleased to inform you that your manuscript has been deemed suitable for publication in PLOS ONE. Congratulations! Your manuscript is now with our production department. 

Kind regards, 

on behalf of

Dr. Silvia Naitza 

Academic Editor

PLOS ONE